# Prime-Boost Vaccination Based on Nanospheres and MVA Encoding the Nucleoprotein of Crimean-Congo Hemorrhagic Fever Virus Elicits Broad Immune Responses

**DOI:** 10.3390/vaccines13030291

**Published:** 2025-03-10

**Authors:** Eva Calvo-Pinilla, Sandra Moreno, Natalia Barreiro-Piñeiro, Juana M. Sánchez-Puig, Rafael Blasco, José Martínez-Costas, Alejandro Brun, Gema Lorenzo

**Affiliations:** 1Centro de Investigación en Sanidad Animal, INIA, CSIC, Valdeolmos, 28130 Madrid, Spain; moreno.sandra@inia.csic.es (S.M.); brun@inia.csic.es (A.B.); 2Centro Singular de Investigación en Química Biolóxica e Materiais Moleculares (CIQUS), Departamento de Bioquímica e Bioloxía Molecular, Universidade de Santiago de Compostela, 15705 Santiago de Compostela, Spain; natalia.bapi@gmail.com (N.B.-P.); jose.martinez.costas@usc.es (J.M.-C.); 3Departamento de Biotecnología, Instituto Nacional de Investigación y Tecnología Agraria y Alimentaria (INIA-CSIC), Ctra. de la Coruña km 7.5, 28040 Madrid, Spain; spuig@inia.csic.es (J.M.S.-P.); blasco@inia.csic.es (R.B.)

**Keywords:** Crimean–Congo hemorrhagic fever virus, nanospheres, modified vaccinia Ankara virus, NP, immune responses

## Abstract

**Background/Objectives**: Crimean–Congo hemorrhagic fever virus (CCHFV) is an emerging, widely distributed zoonotic tick-borne pathogen. The virus causes severe disease in humans, and numerous wild and domestic animals act as reservoirs of it. Unfortunately, there are no effective therapies or safe vaccines commercialized nowadays for this particular virus. As CCHF (Crimean–Congo hemorrhagic fever) is a serious threat to public health, there is an urgent need to investigate the development of safe and effective vaccination strategies further. **Methods**: In this work, we have employed two immunization platforms based on protein nanoparticles and a modified vaccinia Ankara (MVA) viral vector using the nucleoprotein (NP) as the target antigen. The humoral and cellular immune responses were characterized by ELISA, ICS, and cytokine measurement. **Results**: This work shows that a single dose of the vaccine candidates was not as immunogenic as the heterologous vaccination using nanoparticles and MVA. A prime with NP nanoparticles (NS-NP) and a boost with MVA-expressing NP were capable of triggering significant levels of humoral and cellular immune responses against CCHFV in mice. **Conclusions**: Our study shows that the NS-NP/MVA-NP vaccination strategy effectively elicits a robust humoral and cellular immune response in a mouse model, emphasizing its potential as a protective approach against CCHFV lineages.

## 1. Introduction

Crimean–Congo hemorrhagic fever virus (CCHFV) is a vector-borne pathogen affecting humans and livestock that causes severe disease in humans and is classified as a biosafety level 4 (BSL4) agent. This viral infection can lead to severe hemorrhagic fever, with a mortality rate of up to 30–40% observed in certain outbreaks worldwide [1]. The transmission vectors of this tick-borne pathogen are mainly *Hyalomma* species, whose geographical range has increased in the last decade, which highlights the high risk of the emergence of CCHFV in non-endemic regions. CCHFV is present in Asia, the Middle East, Africa, and Eastern and Southern Europe [2], with recent cases emerging in new areas across the Mediterranean [3]. CCHFV was first isolated in 2010 in *H. lusitanicum* ticks that were collected from red deer (*Cervus elaphus*) in Caceres, Spain [4]. Since 2013, a rise in human CCHF clinical cases has been observed in western and southwestern Spain [5,6]. Similarly, CCHFV infections have recently increased in Afghanistan and Iraq, along with a higher mortality rate than in previous years [7,8]. Although CCHFV leads to hemorrhagic symptoms in humans, this virus causes asymptomatic infections in a large diversity of domestic and wild animals, such as cattle, goats, sheep, rodents, and birds, acting as amplifying hosts [9]. The zoonotic transmission from viremic animals to humans is a public health concern, highlighting the urgent need for measures to monitor the circulation of virus. Livestock workers and veterinarians are high risk occupations for CCHF since viremia is documented in sheep, goats, cattle, and camels [10,11]. 

CCHFV belongs to the genus *Orthonairovirus* of the family *Nairoviridae*, order *Hareavirales*, class *Bunyaviricetes* [12]. CCHFV, like all orthonairoviruses, is an enveloped virus containing a tripartite, negative polarity, single-stranded RNA genome. The viral genome consists of three RNA segments referred to as the small (S), medium (M), and large (L) segments. The L RNA segment is the one that encodes the L RNA-dependent RNA polymerase (L-RdRp), while the M segment encodes a polyprotein that is rapidly processed by host cell proteases into surface glycoprotein precursors (GPC). GPC are subsequently cleaved into mature Gn, Gc, and several non-structural proteins, including mucin, GP38, and NSm [13]. Finally, the S segment encodes two proteins: the nucleoprotein (NP), responsible for the encapsidation of the three RNA segments into ribonucleoprotein particles (RNPs), and the non-structural protein (NSs) that is encoded by the positive-sense ORF of the S segment genome [14]. The CCHFV phylogenetic diversity is mainly associated with the three RNA segments due to reassortment events between strains [15]. In this regard, the currently identified CCHFV strains can be classified into seven distinct lineages or clades (Asia-1, Asia-2, Africa-1, Africa-2, Africa-3, Europe-1, and Europe-2) [16].

Currently, a licensed vaccine against CCHF is not available. In order to achieve effective vaccines against CCFHV, several research groups have developed different approaches that are mainly based on viral glycoproteins [17,18,19]. Nonetheless, immune responses against Gn and Gc are highly restricted to the same genotype due to the variability of their sequences [20]. Considering this, the induction of broad-spectrum immune responses against conserved antigens may be the most coherent strategy to confer protective immunity against the different strains and genotypes of CCHFV. In this regard, the amino acid sequence of the NP is highly conserved among CCHFV strains [20], and different antigenic epitopes scattered throughout its sequence have been described [21], making this antigen a good candidate to trigger broad immunity among different CCHFV lineages. The major aim of the present study was to investigate the immunogenicity of the viral NP expressed by two platform systems: nanoparticles derived from the avian reovirus non-structural protein muNS and a modified vaccinia Ankara (MVA) viral vector. These nanoparticles range in diameter between 400 and 500 nm and are called nanospheres (NS). They are very easy to obtain from bacteria by simple mechanical means (Figure 1). The truncated version muNS-Mi has been previously reported to form spherical inclusions that can be fused to any protein of interest as long as it bears the so-called “IC-tag” and both proteins are co-expressed in the same cell [22,23]. Immunogenic proteins immobilized into muNS-Mi inclusions are able to confer protection against other pathogens (as demonstrated in a mouse model), such as different orbiviruses [24]. On the other hand, MVA vectors have shown potential as promising vaccine candidates against several pathogens, including SARS-CoV-2 or Rift Valley fever virus [25,26,27], among others. Here, novel vaccine prototypes against CCHF have been generated, and their use as vaccines has been explored in a mouse model. In light of the results, the heterologous prime–boost immunization triggered strong and solid humoral and cellular immune responses against the NP. In addition, different NP epitopes were found to be strong inducers of CD8^+^ T cell-specific cell-mediated immune responses.

## 2. Materials and Methods

### 2.1. Virus, Bacteria, and Cells

Chicken embryo fibroblast (DF-1) (ATCC CRL-12203) and Vero cells (African green monkey kidney) (ATCC CCL-81) were grown in Dulbecco’s modified Eagle’s medium (DMEM) (Biowest, Nuaillé, France) supplemented with 2 mM glutamine and 10% fetal calf serum (FCS) (Gibco, Waltham, MA, USA). High five (ATCC CRL-10859) insect cells (Trichoplusia ni) were grown in a TC-100 medium (Invitrogen™, Carlsbad, CA, USA) supplemented with 10% FCS. 

MVA-WT and MVA-NP viruses were amplified in DF-1 cells using a MOI of 0.5 following standard protocols. *E. coli* strain XL1-Blue (Stratagene, La Jolla, CA, USA) was used in this work to grow and purify plasmids. Bacteria cells BL21-CodonPlus-RP (DE3) (Agilent Technologies, Santa Clara, CA, USA) were used for protein expression.

### 2.2. Construction of Recombinant Baculovirus to Express NP of CCHFV

The gene sequence of the CCHFV NP (strain ArD39554 from Mauritania) was obtained by PCR amplification from pRB NP plasmid (provided by Dr. Blasco, INIA-CSIC). The forward and reverse primers were 5′-GCGGAATTCATGGAAAACAAAATCGAGGTG-3′ (EcoRI site underlined) and 5′-GCGCTCGAG**TTA**GATAATGTTAGCACTGG-3′ (XhoI site underlined and stop codon in bold), respectively. The amplified PCR product was digested and cloned into pFastBacHT-A plasmid to generate the plasmid pHT-A-NP, containing a 6-histidine tag for further purification of the recombinant protein. Recombinant baculovirus (rBAC) was generated using the Bac-to-Bac system (Invitrogen, Barcelona, Spain) following the supplier’s protocols. rBAC-NP was used to express the recombinant NP CCHFV protein in High five cells. The recombinant protein was purified using the ProBond Purification System (Invitrogen™, CA, USA) by affinity chromatography in nickel chelating resin columns and following standard procedures indicated by the manufacturer for protein purification under native conditions. Purified protein was used by in-house ELISAs to detect specific antibodies.

### 2.3. Production and Purification of muNS-Mi Nanospheres Tagged to NP

First, a plasmid containing the IC-tag sequence was generated, followed by an sHRVp cleavage site. Plasmid pCINeo-muNS was used for PCR amplification with primers 5′-CGGGATCCACCATGGAAGATCACTTGTTGGCTTATCTCAATG-3′ (FW: forward primer; BamHI site is single underlined and ATG initiator is double underlined) and 5′-CCGAATTCAGGCCCTTGAAATAGTACTTCTAGCGCTTCC-3′ (RV: reverse primer; EcoRI site: single underlined; sHRVp site: double underlined). After digestion of the PCR product with restriction enzymes, this was cloned into the plasmid pcDNA 3.1 Zeo to generate the plasmid pcDNA3.1Zeo-IC-sHRVp. Then, the sequence of the CCHFV nucleoprotein was obtained by PCR amplification from the plasmid (pRB NP CCHFV, Dr. Blasco). The FW primer was 5′-GCGGAATTCATGGAAAACAAAATCGAGGTG-3′ (EcoRI site: single underlined), and the RV primer was 5′-GCGCTCGAGTTAGATAATGTTAGCACTGG-3′ (XhoI site: single underlined; stop codon: double underlined). Once digested, the PCR product was cloned into the plasmid pcDNA3.1Zeo-IC-sHRVp to create the plasmid pcDNA-IC-sHRVp-NP. The whole construct was obtained by PCR amplification from pcDNA-IC-sHRVp-NP using the FW primer 5′-GGAGATCTCGCGGAAGATCACTTGTTGGC-3′ (BglII site: single underlined) and RV primer 5′-GCGCTCGAGTTAGATAATGTTAGCACTGG-3′ (XhoI site: single underlined; stop codon: double underlined) and cloned into the MCS2 of the plasmid pET Duet-muNSMi to generate the plasmid pET-Duet-muNSMi/IC-sHRVp-NP. The purification of muNS-Mi nanospheres (NS) carrying the NP was performed as previously described for other nanospheres based on NS in Barreiro-Piñeiro et al.’s work (2018) [28], briefly summarized in Figure 1A,B.

### 2.4. Generation of Recombinant MVA-NP

The nucleocapsid protein of the CCHFV sequence was amplified by PCR from plasmid pRB NP N using forward 5′-GCGGAATTCATGGAAAACAAAATCGAGGTG-3′ (EcoRI site underlined) and reverse 5′-GCGGGATCCTTAGATAATGTTAGCACTGG-3′ (BamHI site underlined and stop codon in bold) primers. The NP sequence was inserted in plasmid pMVA-βGus to generate the pMVA-NP plasmid. The plasmid pMVA-βGus was previously digested with EcoRI and BamHI restriction enzymes. The pMVA-NP transfer plasmid included the F13L gene of MVA and the NP gene under the control of a synthetic early/late promoter. The recombinant MVA-NP virus expressing the NP of CCHFV was produced by infecting DF-1 cells with MVAΔF13L (MOI = 0.1), followed by transfection with the pMVA-NP transfer plasmid using Lipofectamine 3000. Cell cultures were harvested 48 h.p.i. (hours post inoculation), and MVA-NP was isolated and cloned in four rounds of plaque purification.

### 2.5. Purification of the NP from MVA-NP by CsCl Gradient

The NP was purified from MVA-NP to be used as the antigen for ELISAs to detect anti-NP CCHFV IgG in vaccinated groups of mice. DF-1 cells were first infected with MVA-NP (MOI of 1) and then were harvested 48 h post-infection. The cells were frozen at −80 °C, thawed on ice for 10 min, and treated with 600 µL of lysis buffer (10 mM Tris–HCl, pH 7.8, 1.0 mM EDTA, 0.15 M NaCl, 0.25% NP-40) at 4 °C for 30 min. The lysate was centrifuged at 12,000× *g* at 4 °C for 10 min, and the supernatant was collected. This supernatant was carefully layered on top of a 20–50% (*w*/*v*) discontinuous CsCl gradient in Tris-buffered saline, prepared in 1 mL volumes of 20%, 30%, 40%, and 50% CsCl in 5 mL centrifuge tubes. High-speed centrifugation was then performed using a SW55Ti rotor in a BECKMAN COULTER Optima L-100 XP ultracentrifuge at 55,000× *g* at 4 °C for 3 h [29]. Fractions were subsequently collected sequentially from top to bottom (1000 µL each), and fraction 4 was diluted in 4 mL of TBS and centrifuged for 1 h in an SW55Ti rotor at 55,000× *g*. The pellet was collected in 100 µL PBS with a protease inhibitor cocktail, then treated with a 4M urea for solubilization and analyzed by SDS-PAGE and Western blot (WB) in order to analyze the proper presence of the NP.

### 2.6. Immunization of Mice

Mice experiments for this study were carried out in compliance with the European Union guidelines 2010/63/UE and the Spanish Animal Welfare Law 32/2007. The protocols were approved by the Ethical Review Committee at INIA-CISA and the Community of Madrid (Permit number: PROEX 192/17). Female BALB/c mice (Envigo), 8 weeks old, were used in the studies. The mice were housed under pathogen-free conditions and given time to acclimatize to the biosafety level 3 (BSL3) animal facilities at the Animal Health Research Center (INIA-CISA), Madrid, prior to use. Groups of mice (n = 5) were vaccinated intraperitoneally (i.p.) with 50 µg of NS-NP or 10^7^ PFU of MVA-NP on day 21 (single immunization groups 1 and 2). On day 0, a group of mice (n = 5) were vaccinated with 50 µg of NS-NP, and on day 28, animals were boosted with 10^7^ PFU of MVA-NP (group 3). Control mice were inoculated with 50 µg of empty NS and, on day 28, with MVAΔF13L (MVA-WT) (group 4, n = 5).

In order to determine the level of CCHFV NP-specific antibodies triggered after vaccination, sera were collected from all animals on day 42 (two weeks after the boost in prime–boost groups or three weeks after prime in single immunization groups). Mice were sacrificed on day 45, and splenocytes were purified to analyze cellular immune responses specific to the NP.

### 2.7. Enzyme-Linked Immunosorbent Assays

Indirect enzyme-linked immunosorbent assays (ELISA) were performed to measure specific CCHFV NP antibody levels. MaxiSorp plates (Nunc, Rochester, NY, USA) were coated with 300 ng of recombinant NP per well and purified from baculovirus expression system or CsCl gradients from MVA-NP cultures. After washing, the wells were blocked and incubated for 1 h at 37 °C in a 5% milk powder diluted in PBS. Sera collected from mice were diluted at a ratio of 1:50 and analyzed in duplicates. After sera were incubated for 1 h, wells were washed and incubated with an anti-mouse IgG-HRP secondary antibody (Sigma-Aldrich, St. Louis, MO, USA) at a 1:2000 ratio dilution. The reaction was developed with substrate solution TMB (Life Technologies, Carlsbad, CA, USA) and stopped by adding 50 µL of 3N H_2_SO_4_. Results were expressed as optical density (OD) measured at 450 nm in an ELISA reader.

### 2.8. Selection of CCHFV NP Peptides

A selection of class-I restricted H2-K(D)d immunogenic epitopes on the CCHFV NP sequence was performed using the immune epitope database (https://www.iedb.org/). Five peptides (Table 1) were selected on the basis of their highest theoretical affinity score. Those peptides were synthesized to 95% purity in 5 mg/mL batches (GenScript). A peptide concentration of 2.5 µg/mL was used to stimulate splenocytes from the NP-immunized mice and consequently analyze the induced cellular response by intracellular cytokine staining assay (ICS).

### 2.9. Flow Cytometric Analysis

Spleens from mice were collected and homogenized by passing through 70 μm filters. Red blood cells were lysed with a RBC buffer (Sigma-Aldrich, San Louis, MO, USA), and remaining splenocytes were washed and resuspended at 10^7^ cells/mL in RPMI medium (Biowest, Nuaillé, France), supplemented with 10% FCS, 2 mM of L-Glutamine, 100 U of penicillin and 0.1 mg/mL of streptomycin. For the ICS, splenocytes were stimulated with 2.5 µg/mL of NP peptides or an AHSV irrelevant peptide, 4 µg/mL of concanavalin A as a nonspecific stimulus, or left untreated in RPMI 1640 supplemented with 10% FCS. Five hours before the ICS assay, brefeldin A (5 µg/mL) was added to cells to block Golgi transport. At 18 h post-stimulation, cells were washed, stained for surface markers with 0.3 µg of anti-mouse CD3-FITC (Clone REA641, Miltenyi, San Jose, CA, USA) and 0.3 µg of anti-mouse CD8 PerCP-Vio700 (Clone 53-6.7, Miltenyi) and fixed and permeabilized with a PBS 1% FCS-4% formaldehyde-1% saponin buffer. Then, cells were stained intracellularly using an IFN-γ-PE antibody (Clone AN.18.17.24, Miltenyi) at 0.3 µg/well. Data were acquired by a CyFlow Cube 8 (Sysmex Spain S.L., Sant Just Desvern, Spain). The gating strategy used to identify IFN-γ^+^CD8^+^ T-cell populations is shown in Appendix A. Data were analyzed using FlowJo™ v10.0.8 (Tree Star, Ashland, OR, USA).

### 2.10. Immunofluorescence Assays

BL21 CodonPlus-RP (DE3) bacteria were transformed with the dual expression plasmid mentioned in Section 2.3, incubated at 37 °C, and shaked to achieve an OD_600_~0.6–0.8, after which the expression was induced with IPTG (1 mM). After incubation for 3 h, 5 × 10^8^ bacteria/mL were centrifuged and fixed with 4% PFA. Then, bacteria were centrifuged, washed, resuspended in 70% ethanol, and incubated for 1h at RT. After fixation and permeabilization with EtOH, the bacteria were immobilized for 1 h at 4 °C on a chambered cover glass (Cellvis, Mountain View, CA, USA). Glass was precoated with poly-L-lysine (Sigma-Aldrich, San Louis, MO, USA) for 1 h at RT. After three washes with water, the cover glass was air-dried. Bacteria were permeabilized with 25 µg/mL of Lysozyme in TEG (25 mM Tris-HCl pH 8, 10 mM EDTA, 50 mM Glucose) and treated with 50 U/mL of DNase I for 1 h. Bacteria were incubated in a blocking buffer (0.1% BSA and 0.05% Tween 20 in PBS) for 1 h at RT and incubated with primary antibodies (mouse anti-muNS-Mi and rabbit anti-CCHFV11-S NP, 1:500) for 1.5 h. After three washes with PBS, bacteria were incubated with secondary antibodies (anti-mouse IgG Alexa Fluor 488 and anti-rabbit IgG Alexa Fluor in PBS (1 h RT). Images of stained bacteria were obtained with an Andor Dragonfly spinning disk confocal system mounted on a Nikon TiE microscope equipped with a Zyla 4.2 PLUS camera (Andor).

The expression of the NP by recombinant MVA-NP was evaluated by indirect immunofluorescence assays. DF-1 cells were plated on glass coverslips 24 h before, infected with MVA-NP or MVA-WT (MOI of 0.1 and 0.01), and maintained in DMEM containing 2% FCS. At the indicated times, the cells were washed with PBS and fixed/permeabilized with methanol–acetone for 20 min at RT. After washes, cells were incubated with PBS-20% FCS for 1h at RT and then with a polyclonal antibody anti-CCHFV NP (Creative Diagnostics, Shirley, NY, USA) (diluted 1:1000) specific for the CCHFV NP for 1h at RT. After three washes with PBS–FCS 2% (washing and dilution buffer), cells were incubated for 30 min at RT with an anti-rabbit IgG (H+L) secondary antibody conjugated to Alexa fluor 594 (diluted at a ratio of 1:1000 in dilution buffer). DAPI (1:10,000) was used to stain nuclei. The coverslips were washed four times with washing buffer, mounted on glass slides, and analyzed with a Zeiss Axiovert 5 fluorescence microscope (Zeiss, Oberkochen, Germany).

Glass coverslips containing CCHFV-infected Vero cells (provided by Dr. Mirazimi) were used to analyze antibodies in sera from vaccinated animals in an immunofluorescence assay. Diluted sera (1:50) from vaccinated and non-vaccinated mice were incubated overnight at 4 °C. A polyclonal antibody anti-CCHFV-NP (Creative Diagnostics) diluted at a ratio of 1:1000 was used as a positive control. After three washes with PBS–FCS 2%, cells were incubated for 30 min at RT with an anti-rabbit secondary antibody conjugated to Alexa fluor 594 or 488 (1:1000 dilution). The coverslips were washed four times, mounted on glass slides with a mounting medium, and analyzed with a Zeiss Axiovert 5 fluorescence microscope (Zeiss, Oberkochen, Germany).

### 2.11. Measurement of Cytokines by Luminex

Splenocytes from vaccinated and non-vaccinated mice were harvested on day 42 of the experiment and re-stimulated in vitro with the NP peptides (2.5 µg/10^6^ cells) described above. Three days post-restimulation, supernatants from cell cultures were collected in order to perform Luminex assays. Cytokine levels of the samples were analyzed using a multiplex fluorescent bead immunoassay for the quantitative detection of mouse cytokines, in particular the Millipore’s 6-Plex Mouse Cytokine kit that can detect IL12p70, TNF, IL-4, IL-5, IFN-γ, and IL-6 (MILLIPLEX, Burlington, MA, USA). Samples were analyzed with the MAGPIX system (Luminex Corporation, Austin, TX, USA), and data were collected using xPonent software version 3.1. The cytokine concentrations of each sample were calculated using standard curves performed by the software. Values of cytokines that fell below the level of detection of the assay were assigned the lowest detectable concentration as recommended by the manufacturer manual.

### 2.12. Statistical Analyses

Graphical representation and statistical tests of the data collected in this study were performed in GraphPad PRISM version 9.00 (GraphPad Software, San Diego, CA, USA). Differences in antibody levels and immune cell responses between mice groups were calculated by two-way ANOVA with multiple comparisons between groups. P-values lower than 0.05 were considered significant in all cases. The D’agostino–Pearson omnibus normality test was used to check the normal distribution of the data.

## 3. Results

### 3.1. Expression of the IC-Tagged CCHFV NP and Association with muNS-Mi NS

To produce avian reovirus (ARV) muNS-Mi-derived nanospheres (NS) containing a CCHFV NP, a dual plasmid that simultaneously expresses muNS-Mi and an N-terminal IC-tagged NP was generated. The correct expression of both proteins was confirmed by analyzing extracts from bacteria transformed with the dual expression plasmid, either induced with IPTG or not. The Coomassie-stained SDS-PAGE clearly showed the presence of two bands with the apparent molecular weights of muNS-Mi and IC-NP in the IPTG-induced extracts that were not present in the non-induced sample (Figure 1C, compare lanes 5 with 4). WB analysis using anti-NP monoclonal antibody mK1 (Figure 1D, upper lane) or an anti-muNS antibody (Figure 1C, lower lane) confirmed the identity of both proteins. We confirmed the presence of the nanospheres inside the bacteria through TEM and immunofluorescence assay (Figure 1E,F). Dynamic light scattering analysis of the purified sample confirmed the monodisperse characteristics of the purified NS sample.

### 3.2. In Vitro Expression of CCHFV NP by Recombinant MVA-NP

In order to verify the proper expression of the inserted NP CCHFV protein, immunofluorescence assays were carried out in cells infected with MVA-NP and MVA-WT (Figure 2A). Results confirmed the efficient expression of the NP antigen in DF-1 cells. Protein expression was detected at 24 h.p.i. and increased at 48 and 72 h.p.i. DF-1 cells infected with MVA-NP (MOI 1) were used in WB assays (Figure 2B). Using a rabbit polyclonal antibody anti-CCHFV NP (Creative Diagnostics, USA), we detected a band of approximately 52 kDa in cells collected at 24, 48, and 72 h.p.i., consistent with the estimated size of the CCHFV NP.

### 3.3. Induction of Specific Immune Responses After Vaccination with NS-NP and MVA-NP Expressing CCHFV NP

Groups of BALB/c mice were immunized with the different NP constructs in order to evaluate adaptive immune responses against the NP CCHFV protein. We studied the humoral and cellular responses induced in mice immunized with an NS loaded with an NP (NS-NP) and recombinant MVA encoding the same nucleoprotein. In addition, a prime–boost vaccination strategy was performed by combining an NS-NP prime and an MVA-NP boost.

#### 3.3.1. Detection of Specific Humoral Responses After Vaccination

To evaluate whether a CCHFV NP stimulates the induction of specific antibodies, mice were bled on day 42 of the experiment. Humoral immune responses in vaccinated mice were measured by ELISA and immunofluorescence techniques. IgG antibodies against CCHFV NP were quantified by in-house ELISAs using the purified target protein. A homemade indirect ELISA coating with NP was developed, previously produced by recombinant baculovirus or cesium gradient. Figure 3A reveals the purification of the NP by cesium chloride density gradient centrifugation by SDS-PAGE and WB analysis. Figure 3B shows the fractioned elutions of the NP after baculovirus expression and purification by native conditions, indicating that elution 2 reached the highest protein yield, and thus, it was selected for ELISA coating (Materials and Methods). 

A homemade ELISA was employed to analyze NP antibodies in mice vaccinated using different strategies (Figure 4A). After single immunization with NS-NP or MVA-NP, low but detectable levels of NP-specific IgG antibodies were found in mice sera. The highest levels of antibodies specific to NP were detected in mice immunized with the prime–boost NS-MVA-NP, showing significant differences compared with non-immunized mice (Figure 4B).

In addition, antibodies were also detected by immunofluorescence assays employing CCHFV (IbAr 10200) infected Vero cells, whereas in single NS-NP-immunized animals, there was a very weak signal; marked staining was observed in CCHF-infected cells incubated with sera from MVA-NP or NS/MVA-NP vaccinated mice groups (Figure 5). The highest fluorescent signal was detected by using the sera from prime–boost immunized animals. Figure 5 depicts the immunofluorescence images with two representative sera of each vaccinated group, a negative serum from the control NS/MVA group, and a positive commercial serum.

#### 3.3.2. Cellular Immune Responses Induced After NP Vaccination

To determine the ability of the candidate vaccines to elicit CCHFV-specific T-cell responses, flow cytometric analyses were carried out. Three weeks post-immunization (prime only) or two weeks after boosting with MVA (prime–boost groups), spleens of immunized and non-immunized mice were harvested, and splenocytes were stimulated with 2.5 µg/mL of five NP immunodominant peptides (named as peptides 1–5) or with a non-relevant peptide. Thereafter, we determined IFN-γ production in CD8^+^ cells by ICS. Gated CD3 positive events were analyzed for the CD8 marker and IFN-γ production, employing antibodies described in the Materials and Methods. 

The NP peptides elicited induction of specific CD8^+^ T cells in most NP-NS and MVA-NS-immunized mice (Figure 6). The restimulation of cells from control mice with NP peptides or from immunized mice with the irrelevant peptides showed negligible responses. Values obtained with the irrelevant peptides were used to subtract the background. Although detectable specific IFN-γ^+^CD8^+^ responses were observed in immunized animals with only NP or MVA, there were no statistical differences compared to control animals. However, we observed a strong induction of IFN-γ+CD8+ upon restimulation of splenocytes of the NS/MVA-NP-immunized mice with NP peptides, in particular with peptides 1 and 5. Indeed, we observed statistical differences between NS/MVA-NP-immunized animals and the single vaccinated groups (*p* < 0.0001). 

These data show that the heterologous prime–boost immunization with NS/MVA-NP elicits a strong cytotoxic CD8^+^ T cell response in mice, which could be relevant in protection against CCHFV.

#### 3.3.3. Up-Regulation of Cytokines in the Vaccinated Animals

In addition to the cytometry analysis of the cytotoxic immune response, multiplex cytokine analysis was performed using splenocytes obtained 45 days post-vaccination. Isolated cells were stimulated with NP peptides or an irrelevant peptide. Supernatants from stimulated cells were collected after 3 days and used for analyses.

Some cytokine levels increased in vaccinated animals compared to mock immunized animals. In particular, levels of IFN-γ, IL-6, and IL-4 significantly arose in all prime–boost NS/MVA-NP vaccinated mice compared to prime-only vaccinated groups. IL-12p70 cytokine levels also rose in samples from mice subjected to the prime–boost immunization, although no significant differences were observed between groups in this case (Figure 7). Cytokine secretion from prime immunized animals was elevated compared with the control group, although no statistically significant levels were detected.

## 4. Discussion

CCHF is a fatal tick-borne disease with lethality rates of 30% or higher among humans. Increased geographical distribution of this virus is associated with the spread of *Hyalomma* tick vectors, which are being influenced by global warming, bird migration, international trade, and human travel. Although research to develop efficient vaccines against the etiological agent CCHFV is ongoing, no vaccines have been licensed so far [29,30]. A diversity of vaccine approaches have been developed against CCHFV, including inactivated virus (Bulgarian vaccine), viral vector-based vaccines (Adenovirus, MVA, and Vesicular stomatitis virus VSV), subunit vaccines containing CCHFV antigens produced in insect cells or transgenic plant systems, virus-like particles, and nucleic acid-based vaccine platforms (RNA or DNA) [31]. In any case, there are important obstacles to developing a CCHF vaccine, such as the genetic variability of CCHFV, the lack of naturally susceptible animal models, and the requirement to work in a BSL4 laboratory.

Some vaccination approaches have been focused on the development of immune responses against the glycoproteins of the virus [18,32,33] with the induction of neutralizing antibody responses that lead to protection against CCHFV in mice. However, the antibody immune response to CCHFV could not be associated with disease outcome nor protection from vaccines [34]. The usage of the GP38 protein as an antigen is associated with protective responses based on non-neutralizing antibodies in experimental animal models [35]. Considering the high genetic diversity of CCHFV lineages, conserved antigens are important to achieve broad protective immune responses against the virus. In this sense, the CCHFV glycoprotein precursor (GPC) is much less conserved than other viral proteins, with divergent strains exhibiting less than 75% amino acid conservation. In contrast, the NP has approximately 95% or more amino acid similarity between strains [20]. Therefore, the main aim of this study was the development of vaccine approaches based on the NP to trigger broad-spectrum immune responses against CCHFV lineages. Thus, nanospheres from avian reovirus NS and MVA vectors have been used to construct the immunization platforms. These strategies were tested by prime or prime–boost vaccination regimens in mice. There are some advantages in using the proteins immobilized into muNS-Mi nanoparticles, such as (a) presentation of antigens in their native conformation; (b) entry into professional antigen-presenting cells (APCs) by either phagocytosis or receptor-mediated uptake processes; (c) triggering both the exogenous and endogenous pathways, enabling the presentation of viral antigens by MHC class I and class II molecules; (d) no need for adjuvants to enhance protein immunogenicity [36]. Vaccines based on MVA poxvirus vector have demonstrated to be effective against several pathogens, being safe and thermally stable [37], and possess the capability to express large gene inserts. Previous studies have shown that prime–boost immunization strategies are an efficient approach to induce both humoral and cellular immune responses. Indeed, heterologous prime–boost regimes can be more immunogenic than homologous prime–boost strategies, being able to induce both humoral and cellular immunity against a specific antigen using each delivery system individually [36], avoiding the likely pre-existing immunity against the viral vectors used. After an in vivo evaluation of adaptive immune responses, we observed a significant induction of specific antibodies through vaccination with MVA-NP, which increased when NS-NP/MVA-NP prime–boost immunization was conducted. Cellular-mediated immune responses were also analyzed in these studies. As shown by flow cytometric analyses, specific CD8+ T cell immune response was significantly stronger in animals immunized with NS-NP/MVA-NP compared to those immunized with a prime immunization strategy, confirming the advantage of a prime–boost regimen in the stimulating T-cell responses over single vaccinations.

The correlation between vaccine-induced immune responses and protection against CCHFV infection remains under study. The CCHFV envelope glycoproteins are described as the main immunogens for neutralizing antibody responses, although some neutralizing epitopes are not conserved due to the genetic diversity of this virus [38]. Nonetheless, a DNA-based vaccine prototype expressing CCHFV glycoproteins could afford partial protection against a different CCHFV strain [39]. On the other hand, NP triggers high levels of antibodies (non-neutralizing) and T-cell responses and is one of the most conserved proteins of CCHFV [37]. Although they are non-neutralizing antibodies, anti-NP antibodies could be involved in the control of CCHFV infection through TRIM21 (tripartite motif-containing protein 21)-dependent mechanisms in vivo [40]. TRIM21 is an intracellular Fc receptor that binds to all IgG isotypes with high affinity and is responsible for antibody-dependent intracellular neutralization. In addition, previous research showed that cellular immune responses to recombinant CCHFV NP are present in multispecies families [41]. In addition, the evaluation of PBMC responses of CCHF survivors demonstrates that T-cell responses to NP epitopes are detected in greater abundance compared to those directed against CCHFV glycoproteins [42]. Moreover, long-lived CD8^+^ T cell responses to NP are predicted to offer long-term immunity against the virus, responding rapidly to future exposure [43]. Recently, immunization approaches based on NP antigen have been explored, with no complete protection [44] or affording protection in a recent study [45]. Those approaches were based on DNA or MVA, respectively. In the present work, the cloning site and the promoter that controls NP expression by MVA are different from those of Dowall et al. (2016) [44] and allow the production of recombinant protein at high yields. 

Importantly, our studies indicate that specific IFN-γ+CD8^+^ responses are triggered after vaccination with prime–boost NS-NP/MVA-NP. In addition, specific induction of cytokines (IL-6, IL-4, and IFN-γ) was found after restimulation of sera from NS-NP/MVA-NP immunized mice with NP peptides. These results suggest that these potential vaccine approaches direct T-cell responses towards the phenotype Th1. Based on previous works, the induction of high levels of NP antibodies and NP-specific cellular immune responses could protect against viral infection. Our data suggest that the vaccination approach described in this article based on the combination of NS and MVA could be protective against CCHFV. In addition, NS-NP/MVA-NP would trigger a broad protective immunity due to the high amino acid identity of the NP antigen among the different CCHFV clades. 

Some recent vaccine approaches have provided protection in mice with DNA encoding NP and GPC [43] and adenovirus expressing GPC [46]. Another effective immunization strategy based on a viral replicon described high T-cell and antibody responses to the NP (no neutralizing antibodies) [47,48]. However, in the present work, we described for the first time the evaluation of a heterologous prime–boost strategy with nanoparticles and MVA encoding the NP alone, which induces a high level of humoral and cellular immune responses. Among all the variety of vaccine platforms, it is important to choose systems that provide advantages with respect to others. In this respect, vaccine constructs presented in this work are safe, thermally stable, and easy to administer [49]. Moreover, this vaccine candidate allows discrimination between infected and vaccinated animals (DIVA strategy), as serological tests based on the recognition of antibodies specific to glycoproteins are available. This is particularly important when vaccination involves livestock since animal movement restrictions should be applied, as well as correct surveillance studies. 

Our results are consistent with previous works that applied these platforms. A prime dose of NS followed by a boost of MVA triggered stronger humoral and cellular immune responses compared to single vaccinations, being protective against lethal infection with pathogenic orbiviruses [24,50]. Further research will be conducted to improve the induction of specific antibodies after priming with NS-NP, e.g., increasing the NP concentration during vaccination. Nevertheless, the strategy based on nanospheres loaded with NP has been useful for priming the immune system of the animals. Previous studies reinforce the idea that immunogenic nanoparticles are capable of inducing long-lived, antibody-secreting plasma cells and memory B- and T-cells [51]. In addition, recombinant MVA vaccines have been demonstrated to be potent immunogens that can complement other vaccine technologies as a boost to vaccination. MVA is a vector widely used to design vaccines against infectious diseases because of its capability to induce a robust cellular immune response and protection. In addition, heterologous prime–boost regimes are capable of inducing long-term humoral and cellular immune responses against different pathogens [52,53,54]. It has been described that heterologous immunizations can stimulate the immune system in a way that activates various immune cells in a non-specific manner, including those not directly targeted by the vaccine. This activation can amplify the overall immune response, contributing to increased immunogenicity [55].

Considering the CCHFV global distribution scenario, an ideal CCHFV vaccine is bound to be effective against multiple viral lineages. Thus, conserved antigens such as the viral nucleoprotein, NP, are more desirable strategies for stimulating broad immunity. Our results highlight the need for further research to shed light on the potential protective efficacy of the vaccination strategy described in this work. Since the CCHFV agent is classified as a BSL-4 pathogen, challenge experiments require special laboratory facilities with the highest level of safety measures, and because of this, our candidate vaccines have not yet been tested to evaluate their protective capacity. Future in vivo challenge experiments in animal models will be necessary to evaluate whether the NS-NP/MVA-NP immunization has the ability to provide complete protection against the lethal challenge of CCHFV.

## 5. Conclusions

In conclusion, our study demonstrates that the NS-NP/MVA-NP prime–boost vaccination strategy effectively induces a strong humoral and cellular immune response, highlighting its potential as a protective approach against CCHFV lineages. Moreover, we have characterized several NP epitopes that trigger a specific IFN-γ+CD8^+^ T cell response. This approach offers advantages such as safety, thermal stability, and DIVA compatibility. Future research should focus on conducting in vivo challenge experiments to confirm its protective efficacy. 

## Figures and Tables

**Figure 1 vaccines-13-00291-f001:**
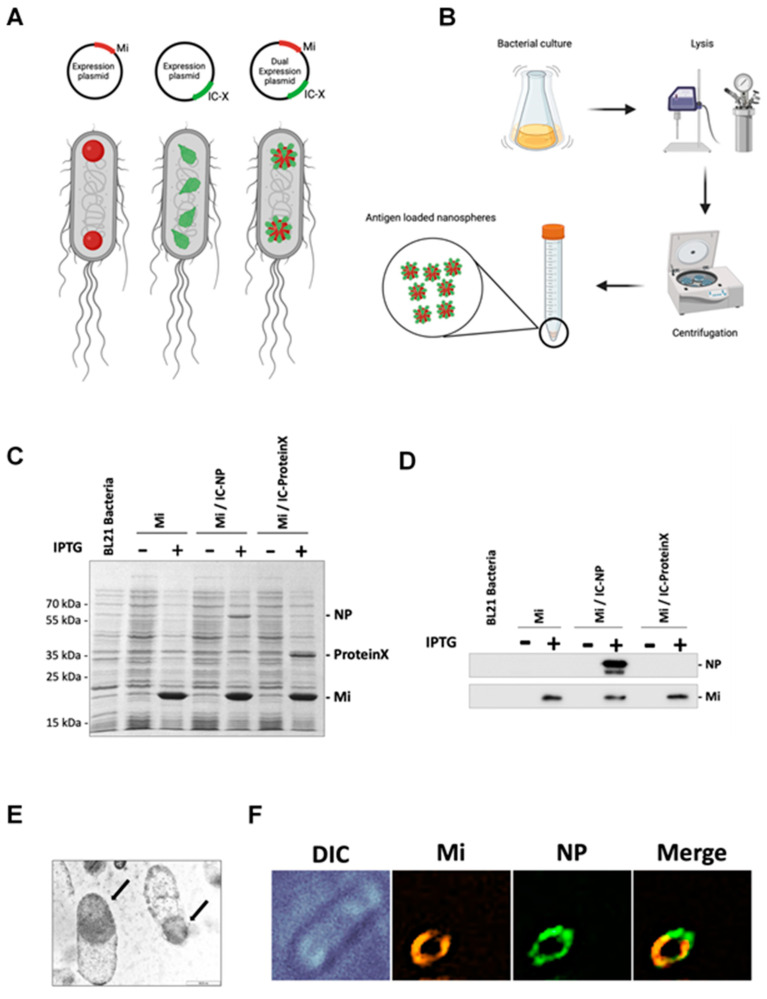
The IC-tagging system and detection of CCHFV NP-loaded NS. (**A**) A schematic representation of the IC-tagging system (patented and published in other methodologies). Viral-derived protein muNS-Mi forms ordered into nanospheres when expressed in any cell (left drawings). A viral-derived tag, called “IC”, fused to the sequence of a desired protein (X) does not change the protein characteristics (middle drawings), but in co-expression, the presence of the IC tag causes the relocation of the tagged proteins to the ordered muNS-Mi nanospheres (right drawings). (**B**) The nanosphere production workflow. The induced culture of transformed bacteria is centrifuged and lysed with mechanical methods. Low-speed centrifugation in a defined buffer produces the NS pellet, which is washed several times to obtain a purified preparation. (**C**) Coomassie-stained SDS-PAGE of extracts from bacteria transformed with muNS-Mi alone (Mi), a dual expression plasmid expressing muNS-Mi and the IC-tagged CCHFV NP (Mi/IC-NP), or a dual expressing plasmid for muNS-Mi and an unrelated protein (Mi/IC-proteinX; see methods). (**D**) WB analysis of the same samples showed in (**A**), conducted using an anti-NP antibody (Figure 1B, upper lane) or an anti-muNS antibody (Figure 1B, lower lane). (**E**) TEM of nanospheres inside bacteria. Bacteria transformed with the dual plasmid for muNS-Mi and the IC-tagged CCHFV NP were induced with IPTG, subsequently fixed, and stained for EM analysis. (**F**) The same sample of bacteria containing CCHFV NP-loaded NS, as shown in E, was analyzed by immunofluorescence using anti-CCHFV11-S nucleoprotein (green) and anti-muNS-Mi (orange). A representative image is shown. DIC: Differential interference contrast microscopy.

**Figure 2 vaccines-13-00291-f002:**
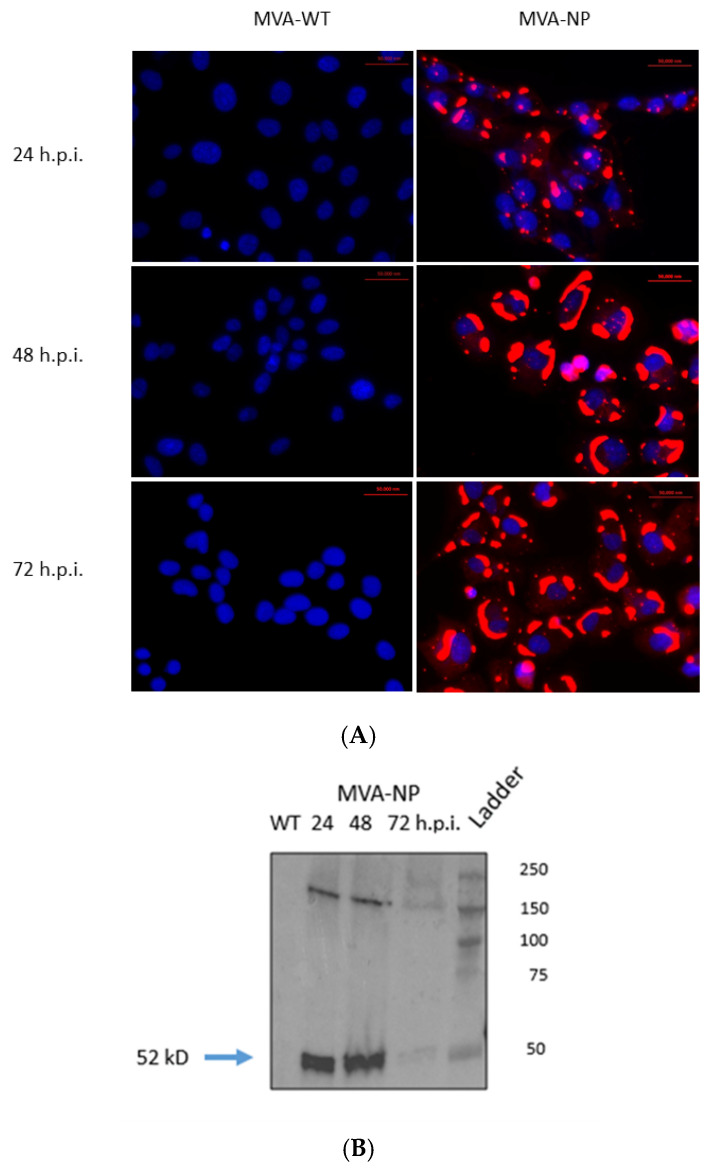
Expression of CCHFV NP by recombinant MVA in DF-1 infected cells. (**A**) DF-1 cells were infected with MVA-WT or MVA-NP (MOI 0.1), and immunofluorescence assays were performed at 24, 48, and 72 h.p.i. A rabbit monoclonal antibody (1:1000) specific for NP CCHFV and an anti-rabbit IgG secondary antibody conjugated to Alexa-594 (1:1000) were used—magnification: 63×. (**B**) DF-1 cells were infected with MVA-NP or MVA-WT at an MOI of 1 and incubated until the cytopathic effect was observed. At 24, 48 or 72 h.p.i., cells were collected and lysed. After SDS-PAGE, WB analysis was performed with an NP CCHFV monoclonal antibody. kD: kilodaltons. The original Western blot figures can be found in Appendix A.

**Figure 3 vaccines-13-00291-f003:**
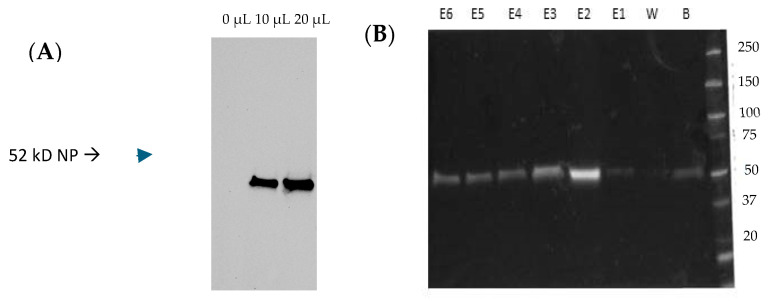
Purification of the CCHFV NP. (**A**) The NP purified from MVA-NP employing a cesium gradient. DF-1 cells infected with MVA-NP (MOI 1) were lysed, and the target protein was purified as described in the Materials and Methods. SDS-PAGE and WB of fraction 4 containing the NP. (**B**) The NP from recombinant baculovirus. High five cells were infected with NP baculovirus at a MOI of 5, and the pellet was processed at 48 h.p.i. The ProBond purification system was used under native conditions following the manufacturer’s manual. B: binding; W: wash; E: elutions. The expected size of the protein NP is 52 kD. kD: kilodaltons. The original Western blot figures can be found in Appendix A.

**Figure 4 vaccines-13-00291-f004:**
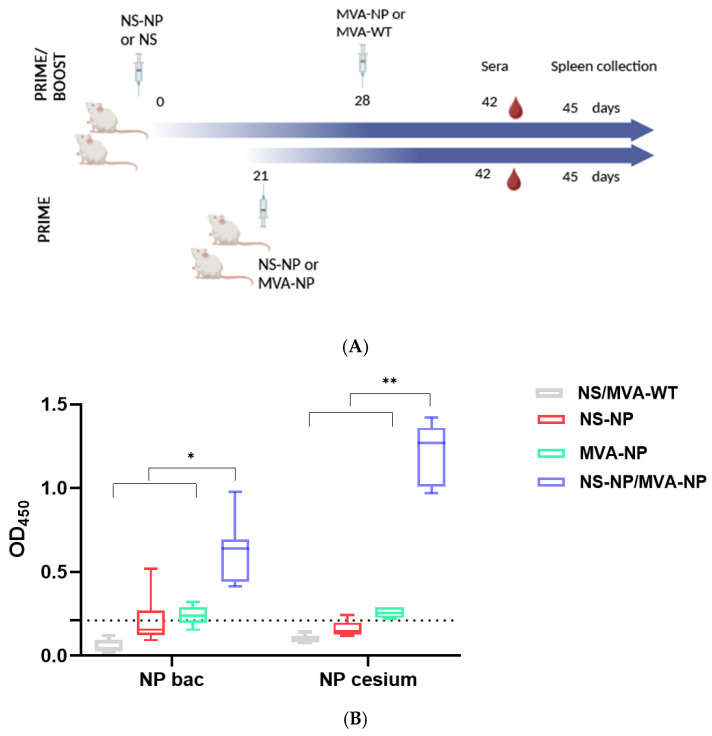
Immunization scheme and humoral responses after vaccination. (**A**) Groups of BALB/c mice (n = 5) were vaccinated as described in the Materials and Methods, and samples were collected for acquired immune response studies. (**B**) Anti-CCHFV NP-specific IgG in mice. Sera were collected from NP-vaccinated and non-vaccinated mice and tested for anti-CCHFV NP-specific by in-house ELISA. Plates were coated with the recombinant NP purified from baculovirus (NP bac) or from MVA-NP by CsCl gradient (NP cesium). Data are represented as boxes of OD_450_ mean values, and error bars show the SD. The estimated cut-off of the assay corresponds to an OD_450_ value of 0.21 (mean of negative + 2SD). Statistical significance: * *p* < 0.05; ** *p* < 0.01.

**Figure 5 vaccines-13-00291-f005:**
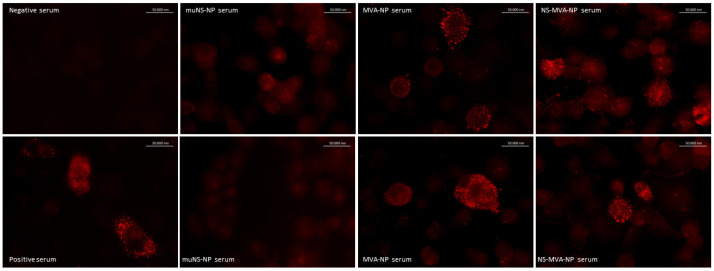
Sera from NP-vaccinated mice recognized cells infected with CCHFV by immunofluorescence assays. Glass coverslips of semi-confluent Vero cells infected with CCHFV were tested by immunofluorescence assay using sera (diluted 1:300) from NP-vaccinated mice (nanospheres and/or MVA), non-vaccinated mice, or a positive polyclonal CCHFV antibody (Creative diagnostics). A secondary Alexa Fluor 594 goat conjugated anti-mouse IgG antibody (red signal) was used to detect the NP in cells—magnification: 63×.

**Figure 6 vaccines-13-00291-f006:**
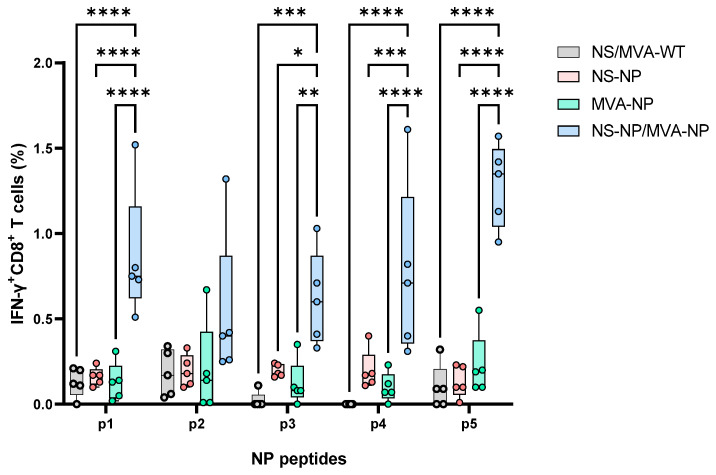
Cellular immune responses against CCHFV in immunized mice. ICS of IFN-γ in CD8^+^ T cells was analyzed by flow cytometry with peptides (p1–p5). On day 45 of the experiment, splenocytes from non-immunized and immunized mice were stimulated with 2.5 µg/mL of predicted immunodominant peptides from NP. The lines in the bars represent the mean values of each group, and the error bars represent the standard error of the mean (SEM). Asterisks denote significant differences between groups (Two-way ANOVA; Tukey multiple comparison tests). * *p*  <  0.05; ** *p*  <  0.002; *** *p*  <  0.001; **** *p*  <  0.0001.

**Figure 7 vaccines-13-00291-f007:**
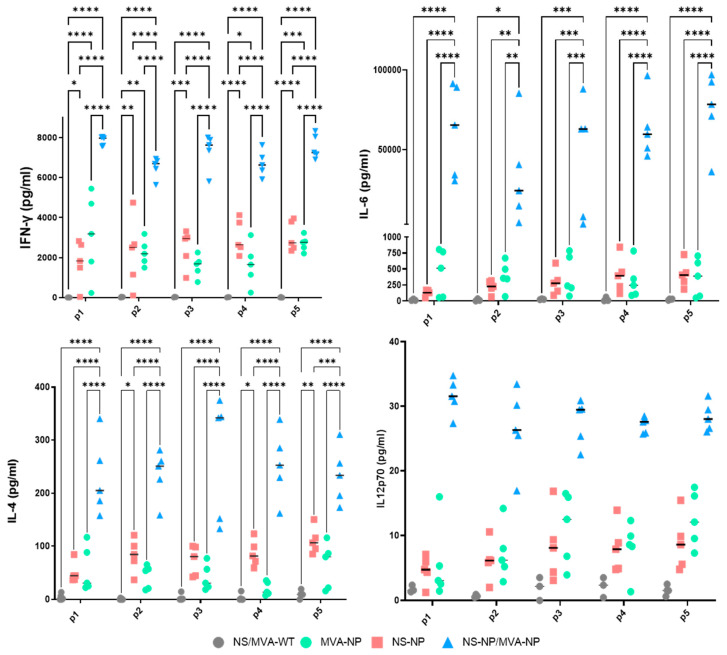
Cytokine secretion levels induced by splenocytes two weeks after prime immunization or after boosting with MVA. Splenocytes from mice were cultured in vitro with the NP peptides (9/10-mer peptide numbered p1 to p5) or an irrelevant peptide. After 72 h, supernatants were collected, and cytokines were determined by Luminex assay (IL-6, IL-4, IL-12p70, and IFN-γ). Values of cytokines from irrelevant peptide-stimulated samples were subtracted from values of NP peptide-stimulated samples. Points in the graphs represent individual values of cytokines for each mouse, and lines represent the mean values of each group of mice. Data were assessed for statistical significance by two-way ANOVA (post hoc Tukey test for multiple comparisons). Asterisks in the graphs denote significant differences between NP CCHFV immunized and control mice. * *p*  <  0.05; ** *p*  <  0.002; *** *p*  <  0.001; **** *p*  <  0.0001. If not indicated, no significance was found.

**Table 1 vaccines-13-00291-t001:** Peptides (9/10-mer peptide sequences) selected among CCHFV NP sequences after class I MHC epitope prediction (https://www.iedb.org/).

Peptide	Position/Size	Sequence
**p1**	140–148/9	RVNANTAAL
**p2**	146–154/9	AALSNKVLA
**p3**	272–280/9	SADSMITNL
**p4**	275–284/10	SMITNLLKHI
**p5**	306–314/9	TAFSSYYWL

## Data Availability

The data presented in this study are available in this article and Appendix A.

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
