# Peer review of "Prime-Boost Vaccination Based on Nanospheres and MVA Encoding the Nucleoprotein of Crimean-Congo Hemorrhagic Fever Virus Elicits Broad Immune Responses"

_vaccines, 2025, doi:10.3390/vaccines13030291_

Round 1
Reviewer 1 Report
Comments and Suggestions for Authors
This manuscript presents the results of development of a new vaccination strategy against Crimean-Congo Hemorrhagic fever virus using a conserved viral protein as an antigen. The authors proposed a heterologous prime-boost vaccination strategy to induce robust NP-specific antibody and T-cell responses, and tested this regimen in BALB/c mice. Although the development of vaccines against CCHFV is an important public health issue, this particular study has several critical flaws that don’t let recommending it for publication in Vaccines.
Major issues:
1. The authors argue that they developed NP-based nanospheres/nanoparticles using the avian reovirus truncated nonstructural protein muNS-Mi, which can form spherical inclusions fused to any protein of interest. However, the authors failed to prove that the expression of these two proteins (NP and muNS-Mi) in bacterial cells indeed form nanoparticles. SDS-PAGE and WB just showed that the correctly folded proteins were expressed, but now evidence of formation of spherical inclusions of NP were presented. The results of immunofluorescent assay (Figure 1C, which lacks the description in the text) are not convincing as there is no information on the size of the spheres, and only one presented particle on the cover slip cannot be treated seriously. The authors should present more solid evidence that the expression of two separate proteins in bacterial cells indeed form nanospheres loaded with the target protein.
2. The design of animal experiment is incorrect. The authors compared single-dose vaccinations with each construct with the two-dose prime-boost vaccination with different vaccines. Mice should have been vaccinated twice with the same vaccines (homologous vaccination), otherwise this comparison with the heterologous prime-boost immunization is incorrect. It is known that two-dose vaccination with homologous antigens always induces more robust antibody and cell-mediated responses than a single-dose immunization. Therefore, there is no proofs that the heterologous prime-boost vaccination is better than administration of two doses of homologous vaccines. Moreover, T-cell responses and analyses of cytokine production after vaccination did not include control (mock) group in the figures, although the authors quote in the text that the vaccinated groups differed from the control group.
3. Since anti-NP antibodies are not neutralizing, their functional activity should be studied to predict any possible mechanism of protection. It is known that NP-specific T-cell responses are protective in a wide range of RNA viruses, but anti-NP antibodies might have ADE effect in some cases, so high anti-NP antibody levels are not always a good idea in designing broadly protective antiviral vaccines.
4. Cell-mediated immune responses were measured only by the proportion of CD8 T cells expressing IFNγ, without distinguishing memory subsets. Currently, a huge number of fluorescent antibodies to various surface markers are available, and determining T-cell memory subsets is a necessary step for assessment of vaccine immunogenicity.
5. The authors claim that the vaccines elicit broad immune responses, but only one NP antigen was used in the immunogenicity analyses.
Minor comments:
1. The Abstract should be re-written, as it mainly presents the introduction on CCHFV and the routes of its distribution. More details on the vaccine approach and the results of the present study should be provided.
2. Lanes 72-73: approach cannot carry glycoprotein.
3. Lane 77: sequences are highly conserved, not identity
4. Lane 82: abbreviations should be deciphered the first time they are used
5. Lanes 91-92: T-cell responses are elicited by the vaccines, not epitopes. This response is epitope-specific.
6. Lane 107: affiliation of Dr. Blasco?
7. Lane 124: “ant”
8. Lane 154: “ClCs”?
9. Lane 169: WB with which antibodies?
10. Lane 176: why BSL-3? Isn’t MVA a BSL-2 pathogen?
11. Lanes 205-206: the response cannot be induced by ICS.
12. Table 1: the title states that the peptides are 9-mers, but there is one peptide which is 10-mer. Peptide 1 is 140-148, not 140-145.
13. Lanes 210-212: the sentence sounds like red blood cells were lysed and then maintained in RPMI.
14. Lane 214: which irrelevant peptide?
15. Lanes 238-239: which secondary antibodies?
16. Lane 263: “cytoquines”
Author Response
Dear Reviewer, thank you for the comments. We have modified the text and included changes to improve the manuscript.
Major issues:
- The authors argue that they developed NP-based nanospheres/nanoparticles using the avian reovirus truncated nonstructural protein muNS-Mi, which can form spherical inclusions fused to any protein of interest. However, the authors failed to prove that the expression of these two proteins (NP and muNS-Mi) in bacterial cells indeed form nanoparticles. SDS-PAGE and WB just showed that the correctly folded proteins were expressed, but now evidence of formation of spherical inclusions of NP were presented.
We agree, it is possible that we cannot show this methodology in detail because we thought that we mentioned several works in the references. The method used, called IC-tagging, has been extensively tested, patented and published (see below) and has been shown to work universally in any protein expression system, either eukaryotic or prokaryotic. The encapsulated proteins are strongly stabilized inside these particles; enzymes work perfectly and even get their pH range extended. Encapsulated particles get co-purified (as is the case presented here) with the simple mechanical method used for NS. As said, all of this characterization has been previously patented and published, and it seems repetitive to us to include this type of information again in this article. However, we have added to figure 1 a schematic representation of how the method works, the co-purification and an EM image showing the presence of these spheres inside all the bacteria where the co-expression of muNS-Mi and CCHFV NP have been induced.
Patents:
1 Patent: Método de producción de la proteína glucosa-6-fosfatasa 2. Inventors: Natalia Barreiro-Piñeiro; Rubén Varela Calviño; Javier Benavente; Jose M. Martinez-Costas. Nº: P201830351. Spain. 09/04/2018.
2 Patent: Protein muNS that can form inclusions in the endoplasmic reticulum, methods for use thereof and uses of same. Inventors: Natalia Barreiro-PIñeiro; Javier Benavente Martínez; José M. Martínez-Costas. Nº: EP158443689.9. European Union. Date: 21/04/2017. Extension USA Nº: US15/513,148. United States of America. Date: 21/03/2017. Extension EU Nº: P201431378. Conferral date: 14/11/2016.
3 Patent: Applications of the protein muNS and the derivatives thereof. Inventors: Alberto Brandariz-Núñez; Rebeca Menaya-Vargas; Javier Benavente; José M. Martínez-Costas. Nº: P201030204. Spain.. Conferral date: 08/11/2011. Extension EU Nº: EP2535348. European Union. Conferral date: 12/2021. Extension USA Nº- US 10,059,745 B2. United States of America. Conferral date: 28/08/2018.
4 Patent: PROTEÍNA DE FUSIÓN muNS CAPAZ DE FORMAR MICROESFERAS Y USOS DE LA MISMA. Inventors: Daniel Abella López, Adrián López Teijeiro, Natalia Barreiro Piñeiro y José M. Martínez Costas. Ref: PCT/ES2024/070125. Date: 01/03/2024.
5 Patent: PROTEÍNA DE FUSIÓN CON INTERDOMINIO MUNS Y MOTIVOS SIM Y SUS USOS. Inventors: Mª del Carmen Rivas Vázquez, María Blaquer Gárate, Natalia Barreiro Piñeiro y José M. Martínez Costas. Ref: P25701ES00 Date: 09/09/2024.
6 Patent: NANOESFERA O MICROESFERA RECUBIERTA FORMADA POR LA PROTEÍNA muNS Y SUS USOS. Inventors: Daniel Abella López, Adrián López Teijeiro, Natalia Barreiro Piñeiro y José M. Martínez Costas Ref: P25455ES00 Date: 25/06/2024
Publications:
1- Daniel Abella-López, Adrián López-Teijeiro, Tomás Pose-Boirazian, Natalia Barreiro-Piñeiro and José M. Martínez-Costas. Novel Nanoencapsulation Platform for Oral Delivery of Peptides: In Vitro Stabilization of AvPAL and Formulation of a Gastrointestinal-Resistant Luciferase. Submitted.
2- Luis Jiménez Cabello; Sergio Utrilla Trigo; Natalia Barreiro Piñeiro; Tomás Pose Boirazian; Jose M Martinez-Costas Costas; Alejandro Marín López. Nanoparticle- and Microparticle-Based Vaccines against Orbiviruses of Veterinary Importance. Vaccines. 10(7):1124. - doi: 10.3390/vaccine, (Spain): 14/07/2022.
5- Alejandro Marín-López; Natalia Barreiro-Piñeiro; Sergio Utilla-Trigo; Diego Barriales; Javier Benavente; Aitor Nogales; José Martinez-Costas; Javier Ortego; Eva Calvo-Pinilla. Cross-protective immune responses against ASHV after vaccination with avian reovirus muNS microspheres and modified Vaccinia virus Ankara. Vaccine. 38, pp. 882- 889. Elsevier, 14/01/2020.
6- Natalia Barreiro-Piñeiro; Irene Lostalé-Seijo; Rubén Varela-Calviño; Javier Benavente; Jose M. Martinez-Costas. Adaptation of the IC-Tagging methodology to express glycoproteins and difficult-to-express membrane proteins: application to the expression of the Type 1 diabetes auto-antigen IGRP. Scientific Reports doi:10.1038/s41598-018-34488-3. 8 - 1, pp. 1 - 12. Nature Publishing Group, 2018.
7- Alejandro Marín-López; Eva Calvo-Pinilla; Diego Barriales; Gema Lorenzo; Javier Benavente; Alejandro Brun; José M. Martínez-Costas; Javier Ortego. Microspheres-prime/rMVA-boost vaccination enhances humoral and cellular immune response in IFNAR(-/-) mice conferring protection against serotypes 1 and 4 of bluetongue virus. Antiviral Research. 142, pp. 55 - 62. Elsevier, 18/03/2017.
8- Marín-López, A.; Otero-Romero, I.; de la Poza, F.; Menaya-Vargas, R.; Calvo-Pinilla, E.; Benavente, J.; Martínez-Costas, J.M.; Ortego, J.. VP2, VP7, and NS1 proteins of bluetongue virus targeted in avian reovirus muNS-Mi microspheres elicit a protective immune response in IFNAR (-/-) mice.Antiviral Research. 110, pp. 42 51. Elsevier, 2014.
9- Alberto Brandariz-Nuñez; Iria Otero-Romero; Javier Benavente; Jose M. Martinez-Costas. IC-tagged proteins are able to interact with each other and perform complex reactions when integrated into muNS-derived inclusions. JOURNAL OF BIOTECHNOLOGY. 155, pp. 284 - 286. Elsevier, 2011.
10- Alberto Brandariz-Nuñez; Rebeca Menaya-Vargas; Javier Benavente; and Jose Martinez-Costas. A Versatile Molecular Tagging Method for Targeting Proteins to Avian Reovirus muNS Inclusions. Use in Protein Immobilization and Purification. PLoS ONE. 5(11) e13961. doi:10 - journal.pone.0013961, PLOS (Public Library of Science), 2010.
11- Alberto Brandariz-Nuñez; Rebeca Menaya-Vargas; Javier Benavente; Jose Martinez-Costas. Avian Reovirus muNS Protein Forms Homo-Oligomeric Inclusions in a Microtubule-Independent Fashion, Which Involves Specific Regions of Its C-Terminal Domain. JOURNAL OF VIROLOGY. 84, pp. 4289 - : 4301. American Society for Microbiology, 2010.
The results of immunofluorescent assay (Figure 1C, which lacks the description in the text) are not convincing as there is no information on the size of the spheres, and only one presented particle on the cover slip cannot be treated seriously. The authors should present more solid evidence that the expression of two separate proteins in bacterial cells indeed form nanospheres loaded with the target protein.
We agree, we made some modifications and add more information. This methodology has been extensively tested before as said. Also, the process of how the IF was performed is described in detail in the “Materials and Methods” section 2.10. “Immunofluorescence assays”. We have extended the information as well in the new figure caption. With all due respect, the reviewer should realize that the size of these particles (300-500 nm) is around the resolution limits of optical microscopy and that this image is quite exceptional showing the simultaneous reaction of two different antibodies (with their secondaries also attached) on such a small particle. The purpose of the image is more illustrative than demonstrative so that, as we have commented before, there is sufficient experimental basis to affirm the co-incorporation of both proteins in the same particle. However, as mentioned above, we have included more information in this regard in the figure, including scale bar for the IF and EM pictures.
- The design of animal experiment is incorrect. The authors compared single-dose vaccinations with each construct with the two-dose prime-boost vaccination with different vaccines. Mice should have been vaccinated twice with the same vaccines (homologous vaccination), otherwise this comparison with the heterologous prime-boost immunization is incorrect. It is known that two-dose vaccination with homologous antigens always induces more robust antibody and cell-mediated responses than a single-dose immunization. Therefore, there is no proofs that the heterologous prime-boost vaccination is better than administration of two doses of homologous vaccines. Moreover, T-cell responses and analyses of cytokine production after vaccination did not include control (mock) group in the figures, although the authors quote in the text that the vaccinated groups differed from the control group.
The reviewer’s opinion is appreciated; however, we have few reasons for not including homologous prime-boost groups. Our group has a great experience in the development of vaccines based on MVA, Adenovirus or DNA plasmid and the study of the immune responses elicited after a homologous or heterologous prime boost using different animal models. Based on our previous results we chose the heterologous prime-boost vaccination strategy because it has demonstrated superior immunogenicity and effectiveness compared to homologous regimens. Moreover, there is a disadvantage to use homologous prime boost based on viral vectors or plasmid and is the generation of immune response against the vaccine vector reducing the effectiveness of the vaccine. This issue made us rethink whether it was necessary a second dose of the same vaccine or better to carry out a second dose using a different virus vector. Nevertheless, we also tested individual vaccines, as single-dose immunization is highly convenient in the vaccine development because it is cost effective and get an effective protection in only one shot.
While other vaccine combinations are possible, we achieved promising results with the strategy we employed. Evaluating additional combinations was not necessary to meet our primary objective, which was to develop a vaccination strategy capable of triggering significant humoral and cellular immune responses against CCHFV.
From an animal ethics perspective, the principle of Reduction requires careful examination of the experimental design to minimize the number of animals used while still ensuring valid results. We do not claim in the paper that double immunizations with NS or MVA would be better or worse than the NS/MVA strategy we employed. Rather, we confirmed that the NS/MVA approach effectively elicits robust humoral and cell-mediated immune responses against CCHFV.
For these reasons, we believe that the groups included in the study are sufficient to address the research objectives.
We added control group in fig 6.
- Since anti-NP antibodies are not neutralizing, their functional activity should be studied to predict any possible mechanism of protection. It is known that NP-specific T-cell responses are protective in a wide range of RNA viruses, but anti-NP antibodies might have ADE effect in some cases, so high anti-NP antibody levels are not always a good idea in designing broadly protective antiviral vaccines.
In the context of this virus, the ADE effect was not described in the bibliography as far as we know. A recent article has described that passive immunization with NP antibodies afforded protection in mice against CCHFV, highlighting the relevant role of NP antibodies.
Leventhal, S.S., Bisom, T., Clift, D. et al. Antibodies targeting the Crimean-Congo Hemorrhagic Fever Virus nucleoprotein protect via TRIM21. Nat Commun 15, 9236 (2024). https://doi.org/10.1038/s41467-024-53362-7
4.Cell-mediated immune responses were measured only by the proportion of CD8 T cells expressing IFNγ, without distinguishing memory subsets. Currently, a huge number of fluorescent antibodies to various surface markers are available, and determining T-cell memory subsets is a necessary step for assessment of vaccine immunogenicity.
In this work, we demonstrated that vaccine strategy afforded the induction of significant immune response at the tested time, compatible with protection comparing with other articles. We agree with the reviewer that it would be very interesting to analyze memory responses and we will do it in future experiments, using memory subsets of antibodies as well as analyzing immune responses after 6 months vaccination.
5.The authors claim that the vaccines elicit broad immune responses, but only one NP antigen was used in the immunogenicity analyses.
As NP antigen from CCHFV is a highly conserved protein among viral isolates, the immune responses triggered against the original strain, would be the same that the one against other CCHFV strains. Then, we use the term broad to refer to this.
Minor comments:
- The Abstract should be re-written, as it mainly presents the introduction on CCHFV and the routes of its distribution. More details on the vaccine approach and the results of the present study should be provided. To reduce the abstract, we moved this sentence to discussion: Increased geographical distribution of this virus is associated with the spread of the Hyalomma tick vectors, influenced by global warming, bird migration, international trade and human travels.
- Lanes 72-73: approach cannot carry glycoprotein. Changed
- Lane 77: sequences are highly conserved, not identity, Changed
- Lane 82: abbreviations should be deciphered the first time they are used. Changed
- Lanes 91-92: T-cell responses are elicited by the vaccines, not epitopes. This response is epitope-specific. Changed
- Lane 107: affiliation of Dr. Blasco? Included
- Lane 124: “ant”. Deleted
- Lane 154: “ClCs”? Changed
- Lane 169: WB with which antibodies? PONER AB Y CASA COMERCIAL
- Lane 176: why BSL-3? Isn’t MVA a BSL-2 pathogen? Our institute (Centro de Investigación en Sanidad Animal) only has animal facility with BSL-3 level, no BSL-2.
- Lanes 205-206: the response cannot be induced by ICS. Changed by : “the induced cellular response”
- Table 1: the title states that the peptides are 9-mers, but there is one peptide which is 10-mer. Peptide 1 is 140-148, not 140-145. Changed.
- Lanes 210-212: the sentence sounds like red blood cells were lysed and then maintained in RPMI. Change by: Red blood cells were lysed with RBC buffer (Sigma-Aldrich, San Louis, MO, USA) and remaining splenocytes were washed and resuspended at 107 cells/mL
- Lane 214: which irrelevant peptide? AHSV peptide
- Lanes 238-239: which secondary antibodies? Anti- IgG-alexa fluor 488 antibody and anti- IgG Alexa fluor 594 antibody.
- Lane 263: “cytoquines”: change by cytokines
Reviewer 2 Report
Comments and Suggestions for Authors
Comment on vaccines-3416695
Title: Prime-boost vaccination based on nanospheres and MVA encoding the nucleoprotein of Crimean-Congo Haemorrhagic fever virus elicits broad immune responses
In this study, the authors employed two immunization platforms based on protein nanoparticles and a modified vaccinia Ankara (MVA) vector using the nucleoprotein of Crimean-Congo hemorrhagic fever virus (CCHFV) as the target antigen. The prime-boost vaccination results demonstrated that this strategy can effectively elicit immune responses. The work is potentially useful. However, the paper was poorly written and presented. Some concerns should be addressed.
Specific issues:
1. The term “broad” in the title was not justified.
2. The Abstract’s background information is excessively detailed; it is recommended to condense this part with a focus on the results.
3. Neutralizing antibodies induced by the candidate vaccines should be examined.
4. Table 1 seems to be useless and it was not mentioned in the main text. Additionally, the naming of the epitopes in the subsequent text is P1, P2, while here it is 1, 2, which should be consistent.
5. A schematic of the vaccine design should be incorporated into Figure 1, which is helpful for better understanding.
6. The markers on the left of Figure 1B need to be labeled.
7. There is an overlapping in Figure 1C.
8. In Section 3.2, Figure 2A should be mentioned before Figure 2B.
9. In Figure 2, there needs a space between "48h" and "72h", no markers were included, and there was no internal control.
10. In Subsection 3.3.1 (Lines 345-349): this part described Figure 3 (protein expression and purification), but the title of Subsection 3.3.1 is Humoral Immune Response, which should be double-checked.
11. Figure 4: Why was a homologous prime-boost immunization group not included?
12. Figure 6: There is only one image in the figure, but the figure legends mentioned A and B.
13. In the Discussion, the authors briefly described the rationale for choosing NP as an immunogen (due to its high conservation) while excluding glycoproteins. However, other studies have used glycoproteins as primary immunogens with favorable immunological outcomes (Wang, Q., Wang, S., Shi, Z., Li, Z., Zhao, Y., Feng, N., Bi, J., Jiao, C., Li, E., Wang, T., Wang, J., Jin, H., Huang, P., Yan, F., Yang, S., & Xia, X. (2022). GEM-PA-Based Subunit Vaccines of Crimean Congo Hemorrhagic Fever Induces Systemic Immune Responses in Mice. Viruses, 14(8), 1664. https://doi.org/10.3390/v14081664). This should be compared and discussed.
Writing suggestions:
1. Line 17: “(200)”?
2. Line 66: “NSs that”?
3. Line 86: “[24][25]” should be modified to “[24, 25]”.
4. Line 87: “Sars-Cov-2” should be modified to “SARS-CoV-2”.
5. Line 92: “T CD8+ specific cell-mediated immune response” should be modified to “CD8+ T cell-specific cell-mediated immune response”.
6. Line 152: “h.p.i.” should be written in full form upon first occurrence.
7. Line 154: “ClCs.” should be modified as “CsCl”.
8. Line 193: “37°Cin” should be modified as “37°C in”.
9. CD8+ and elsewhere should be modified as CD8+.
10. Lines 405 and elsewhere: “CD8+ IFN-γ+ responses” should be modified as “IFN-γ+CD8+ responses”.
11. Lines 425: “vaccinated animals.” should be modified as “the vaccinated animals”.
12. Figure 7C: IL-4 pg/ml should be modified as IL-4 (pg/mL).
13. Line 441: “IFN-y” should be modified as “and IFN-γ”.
14. Line 455: “systems ,” should be modified as “systems,”.

The English writng should be improved significantly.
Author Response
Dear reviewer, thanks for your comments. We have modified the text to improve the paper.
REVIEWER 2:
In this study, the authors employed two immunization platforms based on protein nanoparticles and a modified vaccinia Ankara (MVA) vector using the nucleoprotein of Crimean-Congo hemorrhagic fever virus (CCHFV) as the target antigen. The prime-boost vaccination results demonstrated that this strategy can effectively elicit immune responses. The work is potentially useful. However, the paper was poorly written and presented. Some concerns should be addressed.
Specific issues:
- The term “broad” in the title was not justified. We use broad here because the immune response triggered by NP is broad between lineages due to the conserved aminoacid sequence.
- The Abstract’s background information is excessively detailed; it is recommended to condense this part with a focus on the results. Done.
- Neutralizing antibodies induced by the candidate vaccines should be examined. The NP is an internal protein and antibodies directed against this kind of proteins, may not posse neutralizing activities on their own. Several studies have reported that antibodies targeting CCHFV NP lack neutralizing activity.
- Table 1 seems to be useless and it was not mentioned in the main text. Additionally, the naming of the epitopes in the subsequent text is P1, P2, while here it is 1, 2, which should be consistent. This has been modified and table mentioned in the text.
- A schematic of the vaccine design should be incorporated into Figure 1, which is helpful for better understanding. The figure has been modified
- The markers on the left of Figure 1B need to be labeled. Fig1B is a WB where only the lanes for Ncc and Mi are indicated, that correspond to the positions shown in 1A
- There is an overlapping in Figure 1C. Solved
- In Section 3.2, Figure 2A should be mentioned before Figure 2B. The text has been moved.
- In Figure 2, there needs a space between "48h" and "72h", no markers were included, and there was no internal control. Changed.
- In Subsection 3.3.1 (Lines 345-349): this part described Figure 3 (protein expression and purification), but the title of Subsection 3.3.1 is Humoral Immune Response, which should be double-checked.
The first part of subsection 3.3.1. explains how we purified the proteins to be able to performed a next ELISA assay to test the level of NP antibodies. Then, we explained the analysis of humoral response, as the title of the subsection says.
- Figure 4: Why was a homologous prime-boost immunization group not included?
Our group has a great experience in the development of vaccines based on MVA, Adenovirus or DNA plasmid and the study of the immune responses elicited after a homologous or heterelogous prime boost using different animal models. Based on our previous results we chose the heterologous prime-boost vaccination strategy because it has demonstrated superior immunogenicity and effectiveness compared to homologous regimens. Moreover, there is a disadvantage to use homologous prime boost based on viral vectors or plasmid and is the generation of immune response against the vaccine vector reducing the effectiveness of the vaccine. This issue made us rethink whether it was really necessary if a second dose of the same vaccine or better to carry out a second dose using a different virus vector. Nevertheless, we also tested individual vaccines, as single-dose immunization is highly convenient in the vaccine development because it is cost effective and get an effective protection in only one shot.
While other vaccine combinations are possible, we achieved promising results with the strategy we employed. Evaluating additional combinations was not necessary to meet our primary objective, which was to develop a vaccination strategy capable of triggering significant humoral and cellular immune responses against CCHFV.
From an animal ethics perspective, the principle of Reduction requires careful examination of the experimental design to minimize the number of animals used while still ensuring valid results. We do not claim in the paper that double immunizations with NS or MVA would be better or worse than the NS/MVA strategy we employed. Rather, we confirmed that the NS/MVA approach effectively elicits robust humoral and cell-mediated immune responses against CCHFV.
- Figure 6: There is only one image in the figure, but the figure legends mentioned A and B. We have moved gate strategy from supplementary section to Fig 6A.
- In the Discussion, the authors briefly described the rationale for choosing NP as an immunogen (due to its high conservation) while excluding glycoproteins. However, other studies have used glycoproteins as primary immunogens with favorable immunological outcomes (Wang, Q., Wang, S., Shi, Z., Li, Z., Zhao, Y., Feng, N., Bi, J., Jiao, C., Li, E., Wang, T., Wang, J., Jin, H., Huang, P., Yan, F., Yang, S., & Xia, X. (2022). GEM-PA-Based Subunit Vaccines of Crimean Congo Hemorrhagic Fever Induces Systemic Immune Responses in Mice. Viruses, 14(8), 1664. https://doi.org/10.3390/v14081664). This should be compared and discussed.
We have included more works in the discussion section.
Writing suggestions:
- Line 17: “(200)”? Deleted
- Line 66: “NSs that”? This has been modified.
- Line 86: “[24][25]” should be modified to “[24, 25]”. This has been modified.
- Line 87: “Sars-Cov-2” should be modified to “SARS-CoV-2”. This has been modified.
- Line 92: “T CD8+ specific cell-mediated immune response” should be modified to “CD8+ T cell-specific cell-mediated immune response”. This has been modified.
- Line 152: “h.p.i.” should be written in full form upon first occurrence. This has been modified.
- Line 154: “ClCs.” should be modified as “CsCl”. This has been modified.
- Line 193: “37°Cin” should be modified as “37°C in”. This has been modified.
- CD8+ and elsewhere should be modified as CD8+. This has been modified.
- Lines 405 and elsewhere: “CD8+ IFN-γ+ responses” should be modified as “IFN-γ+CD8+ responses”. This has been modified.
- Lines 425: “vaccinated animals.” should be modified as “the vaccinated animals”.
- Figure 7C: IL-4 pg/ml should be modified as IL-4 (pg/mL). This has been modified.
- Line 441: “IFN-y” should be modified as “and IFN-γ”. This has been modified.
- Line 455: “systems ,” should be modified as “systems,”. This has been modified.
Reviewer 3 Report
Comments and Suggestions for Authors
In this manuscript, Calvo-Pinilla et al describe the immune response elicited by a CCHFV vaccine based on the viral nucleoprotein (NP). Specifically, they use a system they have previously shown to work for other viruses, which consists in two different immunisation platforms: one based on the expression of NP at the surface of a modified vaccinia virus, and the other one based on the expression of NP on protein nanoparticles. Even though there are several papers describing the development of CCHFV vaccines (some also using NP), and that this study does not include challenge experiments, given that there are no currently approved vaccines for CCHFV in humans the presented results are interesting for the field.
While the text is mostly well written (but see below), and the result are well presented, there are a couple aspects the authors should improve on.
Importantly, authors need to clarify the timings of their vaccine strategies, since they are very unclear throughout the text and I believe they are contradictory. In methods (lines 177-181) it should be made clearer that there were two strategies (prime and prime/boost). Additionally, rather than calling “day 21” the vaccination date for the prime strategy, I would call this “day 0”, with sera collected on day 21 and spleen collected on day 24. In any case, I believe the diagram from Fig 4A is contradictory with some statements, e.g.:
- Line 341: “mice were bled at days 26 and 42 post-vaccination”. Shouldn’t this be “mice were bled at days 21 (for prime vaccinated animals) and 42 (from 1st vaccine dose, for prime/boost vaccinated animals)”?
- Line 419: “Three-four weeks after vaccination”. Shouldn’t this be three (for mice vaccinated with the prime strategy) and 6 weeks (in reality, 45 days, for the prime/boost strategy)?
- Line 427: “24 days after the boost vaccination”. Shouldn’t this be 17 days after the boost vaccination?
Additionally, it would have been very informative to compare a homologous prime/boost strategy with the heterologous one. The authors try to address this in the discussion, referring to papers that compare homogeneous and heterogeneous strategies, but it would have been better if they had done the experiment themselves. Since I assume this data is not available, perhaps the authors can describe their results for homologous and heterologous vaccinations using their muNS microspheres/vaccinia approach (references 25 and 47)?
Minor points:
- Line 56: Since the update in bunyavirus classification, arenaviridae are now a family within the class Bunyaviricetes, order Hareavirales. See https://pubmed.ncbi.nlm.nih.gov/39303014/
- Line 110: I cannot see the stop codon in bold?
- Line 199: please provide a subheading title that is more informative than just “peptides”.
- Line 286: please provide some background on the IC tag
- Line 290: describe the positive control Protein X in the text
- Fig 2: swap A and B panels so the order corresponds to the order in the text. Similarly, in the text there is a reference to Fig 4 before a reference to Fig 3. Either swap figures, or remove the reference to Fig 4 on line 334.
- Legends of Fig 2 and Fig 5: replace magnification by a scale bar.
- Legend of Fig 2: these two statements are contradictory: “… and incubated until cytopathic effect was observed. At 24, 48 or 72 h.p.i., cells were collected and lysed.” Was it until cytopathic effect was observed, or at specific hpi?
- Legend of Fig 4: restate this is n=5
- Fig 7D: please show statistical significance
- Lines 495-509: within this paragraph, authors should discuss the role of TRIM21 in the protection by CCHFV NP antibodies (https://pubmed.ncbi.nlm.nih.gov/39455551/). Additionally, they should add other CCHFV NP-based vaccines to their discussion, e.g. https://pubmed.ncbi.nlm.nih.gov/39143104/ and https://pubmed.ncbi.nlm.nih.gov/30857305/
- Please add a reference for the statement in lines 514-515
- Lines 535-537: please provide some details on the viruses against which these vaccines were generated
- The manuscript would benefit from a thorough editing. A non-exhaustive list of grammatical errors/typos includes:
o Line 66: “NSs that” is missing a space.
o Line 79: should be “have been described”
o Line 99: should be “were grown”
o Line 171: should be “Mice experiments for this work were carried”
o Line 210: I would say “and homogenized by passing through 70 µm filters”, or something along this line
o Line 211: “red blood cells were lysed”
o Line 223” should be “is showed in Supplementary Fig. 1”
o Line 410: should be “did not induce”
o Line 495: should be “CCHFV infection remains”
o Line 528: should be “thermally stable and easy”
o Line 532: should be “involves livestock”
o Line 532: movement restrictions of what?
Author Response
Dear reviewer
Thank you for the comments, we have modified the text to improve the manuscript.
REVIEWER 3:
Comments and Suggestions for Authors
In this manuscript, Calvo-Pinilla et al describe the immune response elicited by a CCHFV vaccine based on the viral nucleoprotein (NP). Specifically, they use a system they have previously shown to work for other viruses, which consists in two different immunisation platforms: one based on the expression of NP at the surface of a modified vaccinia virus, and the other one based on the expression of NP on protein nanoparticles. Even though there are several papers describing the development of CCHFV vaccines (some also using NP), and that this study does not include challenge experiments, given that there are no currently approved vaccines for CCHFV in humans the presented results are interesting for the field.
While the text is mostly well written (but see below), and the result are well presented, there are a couple aspects the authors should improve on.
Importantly, authors need to clarify the timings of their vaccine strategies, since they are very unclear throughout the text and I believe they are contradictory. In methods (lines 177-181) it should be made clearer that there were two strategies (prime and prime/boost). Additionally, rather than calling “day 21” the vaccination date for the prime strategy, I would call this “day 0”, with sera collected on day 21 and spleen collected on day 24. In any case, I believe the diagram from Fig 4A is contradictory with some statements, e.g.:
- Line 341: “mice were bled at days 26 and 42 post-vaccination”. Shouldn’t this be “mice were bled at days 21 (for prime vaccinated animals) and 42 (from 1st vaccine dose, for prime/boost vaccinated animals)”?
- Line 419: “Three-four weeks after vaccination”. Shouldn’t this be three (for mice vaccinated with the prime strategy) and 6 weeks (in reality, 45 days, for the prime/boost strategy)?
- Line 427: “24 days after the boost vaccination”. Shouldn’t this be 17 days after the boost vaccination?
We introduced changes in the text to establish the correct dates of sampling and vaccination doses.
Additionally, it would have been very informative to compare a homologous prime/boost strategy with the heterologous one. The authors try to address this in the discussion, referring to papers that compare homogeneous and heterogeneous strategies, but it would have been better if they had done the experiment themselves. Since I assume this data is not available, perhaps the authors can describe their results for homologous and heterologous vaccinations using their muNS microspheres/vaccinia approach (references 25 and 47)?
We have few reasons for not including homologous prime-boost groups. Heterologous prime-boost vaccination strategies have demonstrated superior immunogenicity and effectiveness compared to homologous regimens in multiple studies. Nevertheless, we also tested individual vaccines, as single-dose immunization is highly convenient. Proteins are commonly used as good immunogens to trigger humoral responses and they are administered with adjuvants to improve the antibody response. A second dose of nanospheres could increase the antibody level but is not going to elicit a broad immune response including the cellular response. In the case of two doses of the viral vector, we can trigger the generation of immune response against the vaccine vector reducing the effectiveness of the vaccine. This issue made us rethink whether it was really necessary if a second dose of the same vaccine or better to carry out a second dose using a different virus vector. However, the combination of two approaches is able to elicit strong humoral response and specific cellular responses that can be maintained in the time.
Heterologous vaccinations can lead to a bystander activation, that means that vaccination can stimulate the immune system in a way that non-specifically activates various immune cells, including those not directly targeted by the vaccine. This bystander activation can amplify the overall immune response, contributing to increased immunogenicity (doi.org/10.1186/s10020-021-00317-z).
Minor points:
- Line 56: Since the update in bunyavirus classification, arenaviridae are now a family within the class Bunyaviricetes, order Hareavirales. See https://pubmed.ncbi.nlm.nih.gov/39303014/
Changed
- Line 110: I cannot see the stop codon in bold? Done
- Line 199: please provide a subheading title that is more informative than just “peptides”. Done: Selection of CCHFV NP peptides
- Line 286: please provide some background on the IC tag: More information was added to the Introduction section, and now explicative drawing were added to Figure 1.
- Line 290: describe the positive control Protein X in the text. It has been included in the figure.
- Fig 2: swap A and B panels so the order corresponds to the order in the text. Similarly, in the text there is a reference to Fig 4 before a reference to Fig 3. Either swap figures, or remove the reference to Fig 4 on line 334. Changed.
- Legends of Fig 2 and Fig 5: replace magnification by a scale bar.
Done
- Legend of Fig 2: these two statements are contradictory: “… and incubated until cytopathic effect was observed. At 24, 48 or 72 h.p.i., cells were collected and lysed.” Was it until cytopathic effect was observed, or at specific hpi? This has been changed.
- Legend of Fig 4: restate this is n=5. This has been included.
- Fig 7D: please show statistical significance. There is no statistical significance.
- Lines 495-509: within this paragraph, authors should discuss the role of TRIM21 in the protection by CCHFV NP antibodies (https://pubmed.ncbi.nlm.nih.gov/39455551/). Additionally, they should add other CCHFV NP-based vaccines to their discussion, e.g. https://pubmed.ncbi.nlm.nih.gov/39143104/ and https://pubmed.ncbi.nlm.nih.gov/30857305/
Included in the manuscript.
- Please add a reference for the statement in lines 514-515
- Lines 535-537: please provide some details on the viruses against which these vaccines were generated. Those approaches were based on DNA or MVA, respectively.
- The manuscript would benefit from a thorough editing. A non-exhaustive list of grammatical errors/typos includes:
o Line 66: “NSs that” is missing a space. Changed.
o Line 79: should be “have been described”. Changed.
o Line 99: should be “were grown”. Changed.
o Line 171: should be “Mice experiments for this work were carried”. Done
o Line 210: I would say “and homogenized by passing through 70 µm filters”, or something along this line. Done.
o Line 211: “red blood cells were lysed”. Done
o Line 223” should be “is showed in Supplementary Fig. 1”
o Line 410: should be “did not induce”. Done
o Line 495: should be “CCHFV infection remains”. Done
o Line 528: should be “thermally stable and easy”. Changed by and, are safe and thermally stable
o Line 532: should be “involves livestock”. Changed.
o Line 532: movement restrictions of what? Of Animals.
Reviewer 4 Report
Comments and Suggestions for Authors
This manuscript delves into a crucial area of research: developing safe and effective vaccines against the emerging Crimean-Congo Haemorrhagic Fever Virus (CCHFV), a zoonotic pathogen with increasing global significance and high mortality. The study utilizes two immunization platforms, protein nanoparticles and a modified vaccinia Ankara (MVA) viral vector, targeting the conserved nucleoprotein of CCHFV. The results are promising and potentially groundbreaking, demonstrating significant humoral and cellular immune responses in a murine model.
The manuscript is well-structured and presents data that could significantly advance vaccine development for emerging zoonotic diseases. However, several areas need clarification and additional discussion to improve the overall quality and impact of the work. Below, I outline minor specific issues that should be addressed.
Minor Issues
Results: Schematic Representation of Nanosphere Formation—The manuscript would benefit from a schematic figure illustrating the formation process of the nanospheres. This would provide clarity to readers unfamiliar with the methodology and enhance their understanding of the technology used.
Fluorescence Images—All fluorescence images presented in the manuscript lack scale bars, which are essential for interpreting the spatial dimensions and features of the pictures. Please ensure these are added and appropriately labeled in all relevant figures.
Use of Adjuvants—No adjuvant was used during the immunization. While the manuscript suggests that muNS-Mi nanoparticles may not require adjuvants, a detailed explanation of this choice is warranted. Adjuvants enhance and prolong immune responses, which could be particularly important for long-term immunity. The authors should discuss whether including an adjuvant could improve the immune responses observed and, if so, why it was excluded from this study.
Discussion
Role of Non-Neutralizing Antibodies - In the Discussion section, the authors mention that nucleoprotein (NP) induces high levels of non-neutralizing antibodies. Although these antibodies do not directly prevent infection, they could play a significant role in immune defense by recruiting effector cells to infected tissues. The authors are encouraged to expand on this point by discussing potential mechanisms of action, such as antibody-dependent cellular cytotoxicity (ADCC) or phagocytosis, and how these might contribute to the clearance of infected cells in vivo. This discussion will provide a more comprehensive understanding of the vaccine's effectiveness.
Author Response
Dear reviewer
Thank you for the comments, we have modified the text to improve the manuscript.
REVIEWER 4:
This manuscript delves into a crucial area of research: developing safe and effective vaccines against the emerging Crimean-Congo Haemorrhagic Fever Virus (CCHFV), a zoonotic pathogen with increasing global significance and high mortality. The study utilizes two immunization platforms, protein nanoparticles and a modified vaccinia Ankara (MVA) viral vector, targeting the conserved nucleoprotein of CCHFV. The results are promising and potentially groundbreaking, demonstrating significant humoral and cellular immune responses in a murine model.
The manuscript is well-structured and presents data that could significantly advance vaccine development for emerging zoonotic diseases. However, several areas need clarification and additional discussion to improve the overall quality and impact of the work. Below, I outline minor specific issues that should be addressed.
Minor Issues
Results: Schematic Representation of Nanosphere Formation—The manuscript would benefit from a schematic figure illustrating the formation process of the nanospheres. This would provide clarity to readers unfamiliar with the methodology and enhance their understanding of the technology used. Done, Figure 1.
Fluorescence Images—All fluorescence images presented in the manuscript lack scale bars, which are essential for interpreting the spatial dimensions and features of the pictures. Please ensure these are added and appropriately labeled in all relevant figures.
Scale bars have been included in the images.
Use of Adjuvants—No adjuvant was used during the immunization. While the manuscript suggests that muNS-Mi nanoparticles may not require adjuvants, a detailed explanation of this choice is warranted. Adjuvants enhance and prolong immune responses, which could be particularly important for long-term immunity. The authors should discuss whether including an adjuvant could improve the immune responses observed and, if so, why it was excluded from this study.
Usually, vaccines based on proteins are administered with adjuvants to improve the immune response induced in the host. As we have mentioned in the manuscript, the advantage to use muNS-Mi is that the target protein is incorporated to the nanospheres and can be processed by antigen presenting cells initiating the immune response in the host. Before this work, we carried out a pilot experiment with NP nanospheres using a comitial adjuvant from Seppic. We analyzed the humoral immune response by in house ELISA coating with recombinant N to compare the antibody production between groups. We did not detect any increase in the immune responses, comparing to mice without adjuvant. Thus, we decided not to include adjuvants in the present study.
Discussion
Role of Non-Neutralizing Antibodies - In the Discussion section, the authors mention that nucleoprotein (NP) induces high levels of non-neutralizing antibodies. Although these antibodies do not directly prevent infection, they could play a significant role in immune defense by recruiting effector cells to infected tissues. The authors are encouraged to expand on this point by discussing potential mechanisms of action, such as antibody-dependent cellular cytotoxicity (ADCC) or phagocytosis, and how these might contribute to the clearance of infected cells in vivo. This discussion will provide a more comprehensive understanding of the vaccine's effectiveness.
The nucleocapsid protein (NP) of CCHFV and other bunyaviruses is an immunological target. It is the most abundant protein in the virion and the first synthetized after viral infection. It has been used as a vaccine candidate for several labs with promising results showing protective immunity after viral challenge. It is an internal protein and antibodies directed against this kind of proteins, may not posse neutralizing activities on their own. The role of these anti-NP antibodies is not completely studied. The constant fragment (Fc) of an antibody mediates downstream effector functions via its interaction with Fc-receptors on (innate) immune cells or with C1q, the recognition molecule of the complement system. The interaction with Fc-receptors can lead to killing of virus-infected cells through a variety of immune effector mechanisms including antibody-dependent cell-mediated cytotoxicity (ADCC) and antibody-dependent cellular phagocytosis (ADCP). Antibody-mediated complement activation may lead to complement-dependent cytotoxicity (CDC). In addition, both Fc-receptor interactions and complement activation can exert a broad range of immunomodulatory functions. There are several works focused in this field, anti-NP- antibodies induced by viral infection or after vaccination can bind to viral particles or cells infected forming complexes that can be processed by macrophages, monocytes, natural killer or cytotoxic lymphocytes contributing to the viral clearance. Other recent study suggest that TRIM21 (we included in the text) is involved in the formation of complexes with anti-NP antibodies and virus and they are degraded via proteasome.
Round 2
Reviewer 1 Report
Comments and Suggestions for Authors
The authors revised their manuscript according to the reviewer's comments, which significantly improved its quality. There are several issues that still need to be addressed prior to publication:
- Figure 1C, showing the results of the immunofluorescence assay, shows a whole organism, a bacterium, with nanospheres encapsulated within. But vaccination is not with bacteria, but with purified nanospheres, and my question was how much of the purified protein fraction is structurally consistent with nanospheres? Is there any data on the study of purified nanospheres in this study?
- Figures 4 and 6: put the description of the groups in the same order as they appear in the graph.
- Figure 7: is there a control group in this experiment?
Author Response
Responses Reviewer 1:
The authors revised their manuscript according to the reviewer's comments, which significantly improved its quality. There are several issues that still need to be addressed prior to publication:
- Figure 1C, showing the results of the immunofluorescence assay, shows a whole organism, a bacterium, with nanospheres encapsulated within. But vaccination is not with bacteria, but with purified nanospheres, and my question was how much of the purified protein fraction is structurally consistent with nanospheres? Is there any data on the study of purified nanospheres in this study?
- As we mentioned in the answer to previous review round, the method has been tested extensively before with many different proteins and the specific purification method employed, for example, works only for these particular NS, not for others of the same size. Thus, we currently do routine analysis by optical observation and DLS analysis (As mentioned in section 3.1, line 306-307) that the particles after purification present the usual aspect, size and monodispersion. Also, the SDS-PAGE analysis confirms the protein composition of the purified particles.
- Figures 4 and 6: put the description of the groups in the same order as they appear in the graph.
- This has been changed as requested.
- Figure 7: is there a control group in this experiment?
-We included the control group in the figure. Previously, this was not included to make simpler the graph.
Reviewer 2 Report
Comments and Suggestions for Authors
The revised manuscript has been improved, but needs to revsed by native English speakers.
There are many mistakes thoughout the manuscript.
For example, CCHFV belongs to the genus Orthonairovirus of the family Nairoviridae, order Hareavirales, class Bunyaviricetes.
Comments on the Quality of English LanguageThe manuscript needs to be revised by native English speakers.
Author Response
Responses Reviewer 2: Comments and Suggestions for Authors
The revised manuscript has been improved, but needs to revsed by native English speakers.
An English reviewer from our institution in charge of edition has reviewed the document and made some changes to improve the English quality.
There are many mistakes thoughout the manuscript.
For example, CCHFV belongs to the genus Orthonairovirus of the family Nairoviridae, order Hareavirales, class Bunyaviricetes.
We agree and thank the reviewer and it has been changed as suggested.
Reviewer 3 Report
Comments and Suggestions for Authors
The authors have addressed all my comments.
Author Response
Reviewer 3 agrees with the comments of the authors of round 1. Reviewer 3 has no comments on round 2